# Association between family income to poverty ratio and nocturia in adults aged 20 years and older: A study from NHANES 2005–2010

Yangtao Jia [ID], Jiacheng Ca, Fangzheng Yang, Xinke Dong, Libin Zhou*, Huimin Long*

The Affiliated Lihuili Hospital, Ningbo University, Ningbo, Zhejiang, 315040, People's Republic of China

☯ These authors contributed equally to this work.
* longhuiming@vip.sina.com (HL); zlburo2013@sina.com (LZ)

**Data Availability Statement:** All original data could be publicly available at the NHANES database (https://www.cdc.gov/nchs/nhanes/index.htm). The data and code used for this study are publicly

## Abstract

### Background

Nocturia, the most common lower urinary tract symptom (LUTS), significantly impacts socio-economic factors and individuals' quality of life and is closely related to many diseases. This study utilized data from NHANES 2005–2010 to explore the relationship between family income to poverty ratio (PIR) and the presence of nocturia symptoms in adults aged 20 or older in the United States.

### Methods

Data from the National Health and Nutrition Examination Survey (NHANES) in 2005–2010, including 6,662 adults aged 20 or older, were utilized for this cross-sectional study. The baseline data was used to display the distribution of each characteristic visually. Multiple linear regression and smooth curve fitting were used to study the linear and non-linear correlations between PIR and nocturia. Subgroup analysis and interaction tests were conducted to examine the stability of intergroup relationships.

### Results

Out of the 6,662 adult participants aged 20 or older, 1,300 households were categorized as living in poverty, 3,671 households had a moderate income, and 1,691 households were classified as affluent. Among these participants, 3,139 individuals experienced nocturia, representing 47.12% of the total, while 3,523 individuals were nocturia-free, constituting 52.88% of the total population. After adjusting for all other covariates, it was found that PIR was significantly negatively correlated with nocturia (OR: 0.875, 95%CI: 0.836–0.916 P<0.0001). This trend persisted when PIR was divided into three groups (PIR <1, PIR 1–4, PIR > 4) or quartiles. There was a non-linear negative correlation between PIR and nocturia.

### Conclusion

Our findings indicated that lower PIR was associated with a higher risk of nocturia in adults aged 20 or older in the United States. These findings highlight the importance of considering

available at: https://datadryad.org/stash/dataset/doi:10.5061/dryad.j6q573nnp.

**Funding:** The authors disclosed receipt of the following financial support for this article: This work was supported by the Natural Science Foundation of Ningbo Municipality (2021J281), the Key Cultivating Discipline of LihHuiLi Hospital (2022-P09) and Ningbo Key Clinical Specialty Construction Project (2023-BZZ)

**Competing interests:** The authors have declared that no competing interests exist.

socioeconomic factors in preventing and managing nocturia. Nonetheless, further exploration of the causal nexus between these factors was precluded due to the constraints of a cross-sectional design.

## Introduction

The International Continence Society (ICS) defined nocturia as the condition in which an individual wakes one or more times to void at night [1]. Clinically speaking, nocturia was typically deemed clinically significant if it happened two or more times nightly. Prior studies had associated nocturia occurring more than twice per night with a decline in quality of life [2,3]. Nocturia is the most prevalent and bothersome among all Lower Urinary Tract Symptoms (LUTS), with as many as 60% of elderly individuals (>70 years) and 15–20% of young adults (20–40 years) experiencing voiding two or more times at night [1,4]. Furthermore, nocturia was associated not only with benign prostatic hyperplasia (BPH) [5], obstructive sleep apnea (OSA) [6], coronary artery disease [7], diabetes [8], sleep disorders [6,9] and other conditions, but also with increased all-cause mortality [3]. Therefore, decreasing nocturia could have profound positive effects on quality of life.

The ratio of family income to poverty (PIR) is calculated by dividing family (or individual) income by the poverty threshold specific to each survey year and is an effective measure of income disparity closely related to human life and health [10]. Previous studies showed that lower family income led to poorer health protection, including the inability to access timely and effective preventive and diagnostic measures, which might have led to various diseases [11]. Furthermore, some claimed that PIR was associated with urgent urinary incontinence in adults. As PIR increased, the risk of urinary incontinence significantly decreased [12]. Additionally, research has found that children from lower-income families are more likely to experience bed-wetting at night [13]. Because large-scale cross-sectional studies investigating the relationship between nocturia and PIR levels were scarce, we utilized data from NHANES 2005–2010 to explore a large population of adults aged 20 and above in the United States.

## Materials and methods

### 2.1 Data source

The National Health and Nutrition Examination Survey (NHANES) is a biennial population-based cross-sectional study in the United States that gathers nationally representative data on civilians. This study received approval from the National Center for Health Statistics (NCHS) Research Ethics Review Board, with all participants providing written informed consent. Our analysis focused on participants aged ≥ 20 from three NHANES cycles spanning (2005–2006, 2007–2008, 2009–2010). Data utilized in our study are accessible through the NHANES website (https://www.cdc.gov/nchs/nhanes/index.htm).

### 2.2 Study population and variables

In this survey, we included 31,034 participants from 2005 to 2010, excluding those under 20 years old (n = 13,902), with missing nocturnal voiding data (n = 2,296), missing Poverty Income Ratio (PIR) data (n = 1,087), or meeting specific criteria indicating potential bias (n = 316). Additionally, 7,149 participants had missing data on other covariates. The final sample consisted of 6,662 participants. The participant selection process is outlined in Fig 1.

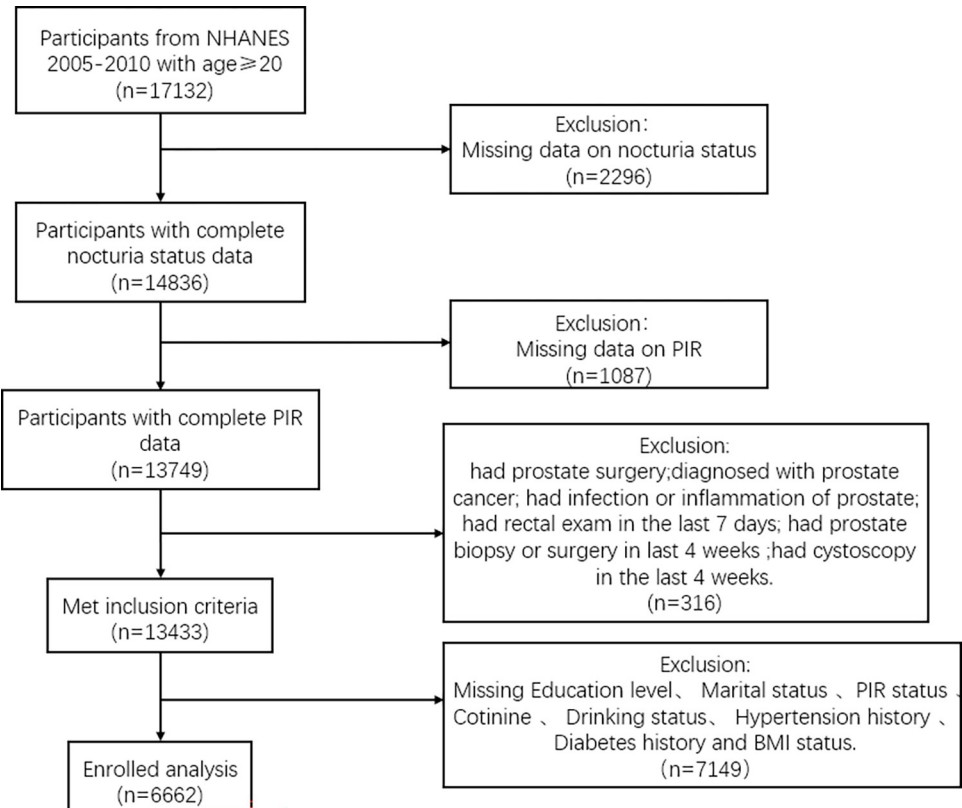

**Fig 1. Flow chart illustrating the selection of participants in NHANES from 2005 to 2010.**

The PIR was the independent variable, with nocturia as the dependent variable. Based on family income and federally recognized poverty levels, PIR was categorized into three groups: poor (PIR <1), middle class (PIR 1–4), and affluent (PIR > 4) [14]. Nocturia was defined as answering "two times" or "two times or more" to a standardized questionnaire item regarding nocturnal voiding frequency.

We explored several variables routinely collected in NHANES, including age, gender, race, education level, marital status, drinking status, BMI, and serum cotinine concentration. Hypertension and diabetes were defined based on standardized questionnaire responses. Hypertension was determined by systolic blood pressure levels $\geq$ 130 mm Hg and/or diastolic blood pressure levels $\geq$ 80 mm Hg or answer "yes" to any of the following questions: "Have you ever been told by a doctor or other health professional that you had hypertension, also called high blood pressure", "Because of high blood pressure/hypertension, have you ever been told to take prescribed medicine?" [15]. Diabetes was defined by glycosylated hemoglobin levels $\geq$ 6.5% and/or fasting blood glucose levels $\geq$ 126 mg/dl, or answer "yes" to any of the following questions: "Have you ever been told by a doctor or health professional that you had diabetes or sugar diabetes?" "Are you now taking diabetic pills to lower your blood sugar?" [16].

## 2.3 Statistical analysis

Baseline characteristics were summarized for continuous and categorical variables, categorized by three levels of PIR. Continuous variables were presented as mean ± standard deviation, while categorical variables were presented as frequencies and percentages. We employed multivariate logistic regression models to examine the linear relationship between nocturia and

PIR, utilizing a weighted smooth curve fitting method to assess non-linear associations. Odds ratios (OR) and corresponding 95% confidence intervals (CI) quantified the strength of association. Model 1 remained unadjusted, while Model 2 adjusted for age, gender, and race/ethnicity. Model 3 additionally adjusted for marital status, education level, serum cotinine concentration, alcohol consumption status, diabetes status, hypertension status, and BMI status. Interaction and stratified analyses were conducted considering the listed covariates. Statistical analyses were performed using STATA v16.0 and R (version 4.2.2). A significance level of $p < 0.05$ was applied.

## Results

### 3.1 Baseline characteristics

We included 6,662 participants from NHANES cycles between 2005 and 2010 and divided the weighted features into three groups based on PIR. (Table 1) The mean age of the participants was 48.35 ± 0.23, with an average PIR of 3.04 ± 024. Compared to low-income individuals (PIR < 1), those with high income (PIR ≥4) were older (PIR < 1: 43.26 years vs PIR ≥4: 48.93 years) and predominantly male (PIR < 1: 43.92% vs PIR ≥4: 53.40%). They were also more likely to be Non-Hispanic White (PIR < 1: 36.23% vs PIR ≥4: 64.64%) and have completed high school or above (PIR < 1: 24.46% vs PIR ≥4: 75.52%). Furthermore, they were more inclined to be married/living with partner (PIR < 1: 47.69% vs PIR ≥4: 74.93%). The number of individuals with serum cotinine concentration >3 was often lower (PIR < 1: 41.85% vs PIR ≥4: 19.04%), while the number of those consuming alcohol more than 12 times a year was higher (PIR < 1: 65.46% vs PIR ≥4: 79.72%).

### 3.2 Relationship between PIR and nocturia

The results of multivariate linear regression analysis in three models indicated a significant negative correlation between PIR and nocturia (Model 1, OR = 0.868, 95% CI: 0.836–0.900; Model 2, OR = 0.862, 95% CI: 0.828–0.897; Model 3, OR = 0.875, 95% CI: 0.836–0.916). Further analysis treating PIR as a categorical variable, with poor individuals (PIR<1) as reference, revealed a significant reduction in the risk of nocturia for the middle class (PIR 1–4) and affluent individuals (PIR>4). When PIR was divided into quartiles and compared with Q1, a significant negative correlation with nocturia risk was observed in Q2, Q3, and Q4. (Table 2) Additionally, we conducted restricted cubic spline regression analysis in Model 3, indicating a non-linear negative correlation between PIR and nocturia. (Fig 2).

### 3.3 Subgroup analysis

Based on all covariates, we found an unstable correlation between PIR and nocturia in the stratified subgroup analysis. Although a negative correlation between PIR and nocturia was observed in the majority of subgroups, the results of interaction tests provided no evidence of an association between this negative correlation and the variables (Fig 3).

## Discussion

This study investigated the relationship between PIR and nocturia symptoms in individuals aged 20 and above. Our model construction and analysis revealed a significant negative correlation between PIR and nocturia symptoms (PR: 0.875; 95% CI: 0.836–0.91; P < 0.0001). Furthermore, constructing restricted cubic spline (RCS) curves unveiled a notable non-linear relationship, indicating a gradual reduction in the risk of nocturia with increasing PIR (P value < 0.0001, P for non-linear = 0.0093). Although subgroup analyses were conducted,

**Table 1. Baseline characteristics of participants stratified by PIR.**

| PIR | <1 | 1–4 | >4 | total | P value |
|---|---|---|---|---|---|
| N | 1300 | 3671 | 1691 | 6662 | |
| PIR | 0.62±0.01 | 2.36±0.18 | 4.84±0.01 | 3.04±0.24 | 0.222 |
| Age (years) | 43.26±0.54 | 49.18±0.34 | 48.93±0.36 | 48.35±0.23 | 0.077 |
| Nocturia | | | | | <0.0001 |
| No | 611(47%) | 1,915(52.17%) | 997(58.96%) | 3,523(52.88%) | |
| Yes | 689(53%) | 1,756(47.83%) | 694(41.04%) | 3,139(47.12%) | |
| Gender | | | | | <0.0001 |
| Male | 571(43.92%) | 1,803(49.11%) | 903(53.4%) | 3,277(49.19%) | |
| Female | 729(56.08%) | 1,868(50.89%) | 788(46.6%) | 3,385(50.81%) | |
| Age | | | | | <0.0001 |
| 20–34 | 377(29%) | 825(22.47%) | 292(17.27%) | 1,494(22.43%) | |
| 35–49 | 332(25.54%) | 820(22.34%) | 438(25.90%) | 1,590(23.87%) | |
| 50–64 | 326(25.08%) | 873(23.78%) | 607(35.90%) | 1,806(27.11%) | |
| ≥65 | 265(20.38%) | 1,153(31.41%) | 354(20.93%) | 1,772(26.60%) | |
| Race | | | | | <0.0001 |
| Mexican American | 355(27.31%) | 716(19.5%) | 153(9.05%) | 1,224(18.37%) | |
| Other Hispanic | 141(10.85%) | 295(8.04%) | 101(5.97%) | 537(8.06%) | |
| Non-Hispanic White | 471(36.23%) | 1,740(47.4%) | 1,093(64.64%) | 3,304(49.59%) | |
| Non-Hispanic Black | 281(21.62%) | 779(21.22%) | 261(15.43%) | 1,321(19.83%) | |
| Other Race | 52(4%) | 141(3.84%) | 83(4.91%) | 276(4.14%) | |
| Education level | | | | | <0.0001 |
| Less than high school | 697(53.62%) | 1,117(30.43%) | 140(8.28%) | 1,954(29.33%) | |
| High school | 285(21.92%) | 1,031(28.08%) | 274(16.2%) | 1,590(23.87%) | |
| More than high school | 318(24.46%) | 1,523(41.49%) | 1,277(75.52%) | 3,118(46.8%) | |
| Marital status | | | | | <0.0001 |
| Widowed/Divorced/ | 680(52.31%) | 1,464(39.88%) | 424(25.07%) | 2,568(38.55%) | |
| Separated/Never married | | | | | |
| Married/Living with partner | 620(47.69%) | 2,207(60.12%) | 1,267(74.93%) | 4,094(61.45%) | |
| Cotinine | | | | | <0.0001 |
| <0.015 | 164(12.62%) | 718(19.56%) | 422(24.96%) | 1,304(19.57%) | |
| 0.015–3 | 592(45.54%) | 2,024(55.13%) | 947(56%) | 3,563(53.48%) | |
| >3 | 544(41.85%) | 929(25.31%) | 322(19.04%) | 1,795(26.94%) | |
| BMI | | | | | 0.024 |
| <25 | 343(26.38%) | 932(25.39%) | 472(27.91%) | 1,747(26.22%) | |
| 25–30 | 422(32.46%) | 1,209(32.93%) | 593(35.07%) | 2,224(33.38%) | |
| ≥30 | 535(41.15%) | 1,530(41.68%) | 626(37.02%) | 2,691(40.39%) | |
| Drink at least 12 drinks/year | | | | | <0.0001 |
| No | 449(34.54%) | 1,210(32.96%) | 343(20.28%) | 2,002(30.05%) | |
| Yes | 851(65.46%) | 2,461(67.04%) | 1,348(79.72%) | 4,660(69.95%) | |
| Hypertension history | | | | | 0.005 |
| No | 533(41%) | 1,372(37.37%) | 702(41.51%) | 2,607(39.13%) | |
| Yes | 767(59%) | 2,299(62.63%) | 989(58.49%) | 4,055(60.87%) | |
| Diabetes history | | | | | <0.0001 |
| No | 901(69.31%) | 2,544(69.3%) | 1,321(78.12%) | 4,766(71.54%) | |
| Yes | 399(30.69%) | 1,127(30.7%) | 370(21.88%) | 1,896(28.46%) | |

**Table 2. Odds ratios for the prevalence of nocturia stratified by PIR.**

|  | Model 1 | P value | Model 2 | P value | Model 3 | P value |
|---|---|---|---|---|---|---|
| **PIR** | 0.868(0.836–0.900) | <0.0001 | 0.862(0.828–0.897) | <0.0001 | 0.875 (0.836–0.916) | <0.0001 |
| **PIR** |  |  |  |  |  |  |
| **<1** | Reference |  | Reference |  | Reference |  |
| **1–4** | 0.81(0.716,0.923) | 0.001 | 0.722(0.632,0.825) | <0.0001 | 0.738(0.642,0.848) | <0.0001 |
| **>4** | 0.617(0.534–0.714) | <0.0001 | 0.572(0.490,0.669) | <0.0001 | 0.627(0.527,0.745) | <0.0001 |
| **PIR** |  |  |  |  |  |  |
| **Q1** | Reference |  | Reference |  | Reference |  |
| **Q2** | 0.830(0.704–0.978) | 0.026 | 0.726(0.612–0.862) | <0.0001 | 0.724(0.607–0.865) | <0.0001 |
| **Q3** | 0.627(0.531–0.740) | <0.0001 | 0.579(0.486–0.690) | <0.0001 | 0.591(0.489–0.713) | <0.0001 |
| **Q4** | 0.564(0.479–0.663) | <0.0001 | 0.528(0.443–0.628) | <0.0001 | 0.564(0.463–0.686) | <0.0001 |

Model 1 adjusts for none.

Model 2 adjusts for age, ethnicity and gender.

Model 3 adjusts for age, gender, ethnicity, education, BMI, marital status, cotinine, alcohol status, Hypertension history and Diabetes history.

Q1: PIR≤1.15, Q2:1.15≤PIR<2.14, Q3: 2.14≤PIR<4.08, Q4: PIR>4.08.

interaction results indicated that various covariates were not significantly dependent on the relationship between PIR and nocturia.

Nocturia was a prevalent chronic condition associated with various ailments, such as cardiovascular diseases, endocrine disorders, chronic respiratory diseases, and mortality, frequently leading to a considerable decline in quality of life [17–21]. Nevertheless, existing treatment and prevention methods were limited, yielding unsatisfactory management

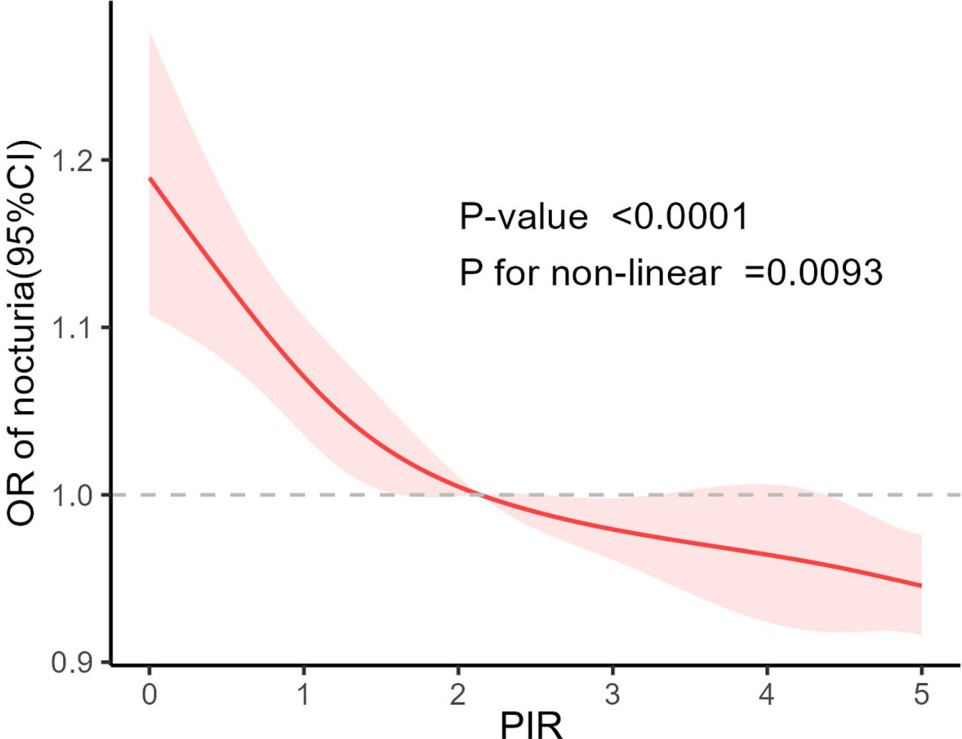

**Fig 2. Restricted cubic splines showing the relationship between PIR and the presence of nocturia.**

| Characteristic | Total(N) | OR(95%CI) | | P value | P for interaction |
|---|---|---|---|---|---|
| Gender | | | | | 0.407 |
| Male | 3277 | 0.90(0.85 - 0.94) | | <0.0001 | |
| Female | 3385 | 0.87(0.83 - 0.92) | | <0.0001 | |
| Age | | | | | 0.179 |
| 20-34 | 1494 | 0.85(0.79 - 0.92) | | 0.007 | |
| 35-49 | 1590 | 0.85(0.80 - 0.91) | | 0.016 | |
| 50-64 | 1806 | 0.86(0.81 - 0.91) | | <0.0001 | |
| ≥65 | 1772 | 0.99(0.92 - 1.06) | | 0.164 | |
| Race | | | | | 0.729 |
| Mexican American | 1224 | 0.95(0.87 - 1.04) | | 0.037 | |
| Other Hispanic | 537 | 0.93(0.83 - 1.04) | | 0.342 | |
| Non-Hispanic White | 3304 | 0.88(0.84 - 0.92) | | <0.0001 | |
| Non-Hispanic Black | 1321 | 0.85(0.79 - 0.92) | | <0.0001 | |
| Other Race - Including Multi-Racial | 276 | 0.84(0.72 - 0.97) | | 0.066 | |
| Education | | | | | 0.566 |
| Less than high school | 1954 | 0.89(0.83 - 0.96) | | 0.001 | |
| High school | 1590 | 0.83(0.77 - 0.90) | | 0.001 | |
| More than high school | 3118 | 0.90(0.86 - 0.95) | | <0.0001 | |
| Marital status | | | | | 0.198 |
| Widowed/Divorced/Separated/Never married | 2568 | 0.85(0.80 - 0.90) | | <0.0001 | |
| Married/Living with partner | 4094 | 0.90(0.86 - 0.94) | | <0.0001 | |
| Cotinine | | | | | 0.193 |
| <0.015 | 1304 | 0.96(0.89 - 1.03) | | 0.128 | |
| 0.015-3 | 3563 | 0.88(0.84 - 0.92) | | <0.0001 | |
| >3 | 1795 | 0.83(0.78 - 0.89) | | <0.0001 | |
| BMI | | | | | 0.539 |
| <25 | 1747 | 0.90(0.84 - 0.96) | | 0.002 | |
| 25-30 | 2224 | 0.86(0.81 - 0.91) | | <0.0001 | |
| ≥30 | 2691 | 0.90(0.85 - 0.95) | | 0.005 | |
| Drink at least 12 drinks/year | | | | | 0.45 |
| No | 2002 | 0.87(0.82 - 0.93) | | 0.001 | |
| Yes | 4660 | 0.89(0.85 - 0.92) | | <0.0001 | |
| Hypertension history | | | | | 0.66 |
| No | 2607 | 0.89(0.84 - 0.93) | | 0.002 | |
| Yes | 4055 | 0.88(0.84 - 0.92) | | <0.0001 | |
| Diabetes history | | | | | 0.622 |
| No | 4766 | 0.88(0.84 - 0.92) | | <0.0001 | |
| Yes | 1896 | 0.90(0.84 - 0.96) | | 0.007 | |

0.8   0.9   1.0

**Fig 3. Subgroup analysis of the association between family PIR and nocturia.**

outcomes for nocturia [1]. Prior studies identified several nocturia-related factors, including reduced bladder volume, sleep disturbances, and nocturnal polyuria [22–24]. Nevertheless, as research on nocturia advanced, some studies revealed a significant association between economic income and nocturia. Consequently, we conducted a study utilizing the NHANES database to investigate the relationship between PIR and nocturia. Our results aligned with prior findings suggesting that lower household income was often linked with more adverse effects, and there existed an inverse correlation between PIR levels and nocturia. For instance, a recent cross-sectional study conducted among the Danish population corroborated our findings, indicating a significant decrease in the risk of nocturia among individuals with higher household incomes [25]. Additionally, previous studies had underscored obesity, hypertension, and diabetes as pivotal risk factors for nocturia, aligning with our findings [8,26,27]. Building upon these findings, we postulated that the Household PIR could serve as a potential indicator reflecting the overarching state of nocturia. Moreover, as a measure of economic status, PIR may be a helpful tool in assessing socioeconomic disparities, thus offering a valuable reference point for forecasting the likelihood of nocturia occurrence across diverse demographic groups.

Although the precise mechanism by which PIR influences nocturia remained elusive, we offered potential explanations grounded in existing research. Epidemiological studies indicated that low-income families faced environments detrimental to physical health, struggled with healthy food affordability, and lacked adequate medical coverage [28,29]. Additionally, lower family income correlated with poorer family health protection, hindering timely disease prevention, diagnosis and treatment, thus fostering disease occurrence and progression [11,30]. For instance, women with lower family incomes discussed urinary incontinence with doctors less frequently compared to their higher-income counterparts [31]. Conversely, individuals with higher family incomes could promptly intervene in diseases through early-stage examinations, inhibiting disease progression. [32]. Furthermore, individuals of lower socioeconomic status often contended with heightened stress hormone levels due to various social and psychological factors [33,34]. Stress, as shown in previous studies, played a significant role in the pathophysiology of overactive bladder (OAB) and urge urinary incontinence (UUI), both impacting urinations [34,35]. Considering these points, we asserted that the influence of income level on nocturia risk was plausible. The household PIR could reflect the comprehensive state of nocturia, with a discernible decrease in the risk of nocturia occurring as PIR increases. This observation holds significant importance for the clinical management of nocturia, as it suggests that PIR could be employed as a potential indicator for assessing the risk of nocturia. By evaluating patients' family income status, one can forecast the likelihood of nocturia and intervene with educational measures and treatment in advance to prevent its onset and mitigate its progression.

A significant strength of our study lies in establishing a nationally representative cohort, bolstering the generalizability of our findings. This analysis marked the first attempt to explore the relationship between nocturia and PIR based on the NHANES database. Additionally, our study identified a negative association between household PIR and nocturia, which carried significant clinical implications. Moreover, we controlled for various demographic, socioeconomic, and disease-related variables within the NHANES dataset. Subgroup analyses were performed, stratified by participant characteristics, to assess the impact of different factors on the association between household PIR and nocturia.

Nevertheless, our study possesses certain limitations. Firstly, owing to its cross-sectional design, causal relationships, and underlying mechanisms could not be deduced. Secondly, the diagnosis of nocturia relied on subjective self-reported questionnaire data, which might not have accurately reflected actual conditions and lacked detailed information on nocturnal voiding patterns, potentially introducing information bias. Furthermore, our sample size was

relatively small, possibly introducing selection bias. Additionally, despite accounting for numerous potential confounders and exclusion criteria, other unaccounted confounding factors might have affected the experimental outcomes, potentially introducing confounding bias. Considering these limitations, further prospective or longitudinal studies was needed to validate our findings. Furthermore, larger-scale studies, such as those that included more samples and covariates, would have been necessary to obtain more accurate results.

## Conclusion

We have established the correlation between PIR and the incidence of nocturia. An elevation in PIR correlates with a reduced likelihood of experiencing nocturia, indicating its potential utility as an indicator for predicting the probability of developing nocturia. Despite offering specific clinical insights, the precise underlying mechanisms remain elusive. Therefore, further randomized controlled trials are warranted to furnish additional evidence for preventing and screening nocturia within the populace.

## Acknowledgments

We extend our gratitude to all participants in the US National Health and Nutrition Examination Survey (NHANES) and value the unrestricted access offered to the public by the NHANES database.

## Author Contributions

**Funding acquisition:** Huimin Long.

**Methodology:** Yangtao Jia, Libin Zhou.

**Software:** Yangtao Jia, Fangzheng Yang, Xinke Dong.

**Supervision:** Huimin Long.

**Validation:** Fangzheng Yang.

**Writing – original draft:** Yangtao Jia, Jiacheng Ca, Xinke Dong.

**Writing – review & editing:** Libin Zhou, Huimin Long.

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
