## [Decision Letter · Decision Letter 0]

1 Apr 2024

PONE-D-24-06446Association between family income to poverty ratio and nocturia status: a study from NHANES 2005-2010PLOS ONE

Dear Dr. Jia,

Thank you for submitting your manuscript to PLOS ONE. After careful consideration, we feel that it has merit but does not fully meet PLOS ONE’s publication criteria as it currently stands. Therefore, we invite you to submit a revised version of the manuscript that addresses the points raised during the review process.

We look forward to receiving your revised manuscript.

Kind regards,

Sairah Hafeez Kamran, PhD

Academic Editor

PLOS ONE

“The authors disclosed receipt of the following financial support for this article: This work was supported by the  Natural Science Foundation of Ningbo Municipality (2021J281), the Key Cultivating Discipline of LihHuiLi Hospital (2022-P09) and Ningbo Key Clinical Speciality Construction Project (2023-BZZ) X”

Additional Editor Comments:

Please ensure that the manuscript meets the journal requirements. 

Submission Guidelines | PLOS ONE

Reviewers' comments:

Reviewer's Responses to Questions

**Comments to the Author**

1. Is the manuscript technically sound, and do the data support the conclusions?

Reviewer #1: Yes

Reviewer #2: Yes

Reviewer #3: Yes

2. Has the statistical analysis been performed appropriately and rigorously? 

Reviewer #1: Yes

Reviewer #2: Yes

Reviewer #3: Yes

3. Have the authors made all data underlying the findings in their manuscript fully available?

Reviewer #1: Yes

Reviewer #2: Yes

Reviewer #3: Yes

4. Is the manuscript presented in an intelligible fashion and written in standard English?

Reviewer #1: Yes

Reviewer #2: Yes

Reviewer #3: Yes

5. Review Comments to the Author

Reviewer #1: This article discusses an important topic in the field of public health. It is well written. The data was appropriately analyzed and results were well presented and discussed. Relevant literature was covered.

Reviewer #2: a) Please provide more details on the specific NHANES survey cycles used and the handling of missing data.

b) The rationale for the categorization of PIR into three groups (PIR <1, PIR 1-4, PIR > 4) should be better explained and supported by relevant literature or guidelines.

c) The authors should consider discussing the potential clinical implications of their findings, such as the potential use of PIR as a risk marker for nocturia or the importance of addressing socioeconomic disparities in the management of nocturia.

Overall, the manuscript presents an interesting and well-conducted analysis of the relationship between PIR and nocturia using a nationally representative dataset. With some additional clarifications and discussions, the manuscript could provide valuable insights into the socioeconomic determinants of nocturia and inform future research and clinical practice in this area.

Reviewer #3: ASSOCIATION BETWEEN FAMILY INCOME TO POVERTY RATIO AND NOCTURIA STATUS: A STUDY FROM NHANES 2005-2010

The manuscript titled, “Association between family income to poverty ratio and nocturia status: a study from NHANES 2005-2010”. Overall, using data from NHANES, the publication presents a study examining the association between the occurrences of nocturia in adults aged 20 and above and the Poverty Income Ratio (PIR). This is a criticism based on the given standards.

1. Originality Of Research:

The study aims to answer a particular research question on how wealth inequality affects nocturia, which is a common health problem. Utilizing NHANES data to investigate this relationship seems like a novel method that adds to the existing body of knowledge available on the topic. It was well drafted and the results are original but I suggest a modification in the title to include the age range utilized in the study.

2. Results Publication Status:

The results have not been published elsewhere. A closely related article assessed the “Association of socioeconomic status and overactive bladder in US adults” has been done. The work complies with the criteria of presenting unique research findings because it does not mention that the results have been published anywhere previously.

3. Technical Standard of Experiments and Analyses

The analysis conducted is detailed since it goes to the extent of stratifying to know the effects the study variables on the outcome of the dependent variable. The association between poverty ratio (PIR) and nocturia is examined using multivariate logistic regression models in the study, which account for several variables. Replication of the procedures and statistical analysis is possible because of the detailed descriptions provided.

4. Data Supported Conclusion:

The results of the statistical analysis validate the inferences made from the data. Regardless of the model or variable categorization, there is a constant negative correlation between PIR and nocturia.

5. Clarity Of Language

The background, methods, results, and discussion are all presented in a structured manner. The manuscript clearly conveys the study's findings and is written in Standard English. The conclusions are presented in appropriate fashion.

6. Ethical Standards of Research

Approval of the NHANES study by the Research Ethics Review Board of the National Center for Health Statistics (NCHS) is mentioned in the manuscript. It also says that written informed consent was given by each participant.

7. Adherence To Reporting Guidelines and Data Availability

The manuscript adheres to reporting guidelines by clearly outlining the methods, results, and discussion sections. Data availability is ensured through the utilization of publicly accessible NHANES data. Overall, the manuscript meets the specified criteria for critique. It provides valuable insights into the relationship between poverty and the prevalence of nocturia, contributing to the understanding of socioeconomic factors influencing health outcomes.

Suggestions for improvement however, will be to include further discussion on potential mechanisms underlying the observed correlation and addressing limitations such as the cross-sectional nature of the study and the need for additional prospective research.

Additionally, clarification on potential biases and their mitigation strategies would enhance the manuscript's robustness. From the manuscript, it adheres to appropriate reporting guidelines and therefore suitable for publication with minor changes.

6. PLOS authors have the option to publish the peer review history of their article (what does this mean?). If published, this will include your full peer review and any attached files.

Reviewer #1: **Yes: **Abubakr Abdelraouf Alfadl

Reviewer #2: **Yes: **ISAAC OKOH ABAH

Reviewer #3: No

---

## [Author Response · Author response to Decision Letter 0]

8 Apr 2024

Review Comments 

Reviewer 1: 

Question 1：This article discusses an important topic in the field of public health. It is well written. The data was appropriately analyzed and results were well presented and discussed. Relevant literature was covered.

Our response：We have noted that reviewer 1 has not put forward any questions that need to be revised. We sincerely appreciate the thorough review and acknowledgment of my paper by the first reviewer. 

Reviewer 2

Question 1: Please provide more details on the specific NHANES survey cycles used and the handling of missing data.

Our response: We listed three specific NHANES cycles used in the literature (line 89) and conducted a detailed statistical analysis of missing data (Figure 1).

Question 2: The rationale for the categorization of PIR into three groups (PIR <1, PIR 1-4, PIR > 4) should be better explained and supported by relevant literature or guidelines.

Our response: We classified PIR based on the following references (DOI: 10.3389/fpubh.2022.873805, 10.3389/fendo.2023.1160625 and 10.3389/fonc.2023.1265356). Additionally, eligibility for subsidies under the Patient Protection and Affordable Care Act was used to assess the middle- and high-income groups.

Question 3：The authors should consider discussing the potential clinical implications of their findings, such as the potential use of PIR as a risk marker for nocturia or the importance of addressing socioeconomic disparities in the management of nocturia.

Our response: We further explored the potential utility of PIR as a risk marker for nocturia (line 227-234) and the significance of socioeconomic disparities in managing nocturia (line 206-210).

Reviewer 3

Question 1: A modification in the title to include the age range utilized in the study.

Our response: We revised the title to "Association between family income to poverty ratio and nocturia in adults aged 20 years and older: a study from NHANES 2005-2010" using " in adults aged 20 years and older " to denote the age range (line 2). 

Question 2: Further discussion on potential mechanisms underlying the observed correlation and addressing limitations such as the cross-sectional nature of the study and the need for additional prospective research.

Our response: We further discussed the potential mechanisms underlying the correlation between nocturia and PIR, analyzing its practical application value in clinical relevance (line (line 227-234). Finally, we proposed the need for additional prospective and longitudinal studies to validate our experiments' conclusions and overcome our research's limitations (line 253-256).

Question 3: Clarification on potential biases and their mitigation strategies would enhance the manuscript's robustness.

Our response: We further explored the possibility of bias. Mitigation strategies such as conducting further longitudinal and prospective studies and more extensive scale studies are also proposed (line245-256).

Editorial Formatting Comments

Question 1: Include the number and proportion of the sample with nocturia.

Our response: In the results section of the abstract, we included information regarding the number and proportion of the sample with nocturia (line 38-42). 

Question 2: Include the test statistics.

Our response: In the results section of the abstract, we added the p-value of the correlation between PIR and nocturia in Model 3 (line 44).

Question 3：The conclusion of the abstract could be improved by making it more concise and focused on the key findings and implications of the study.

Our response：We revised the conclusion section of the abstract based on the feedback provided in the attachment (line 48-52).

Question 4: Explain the rationale for using three models.

Our response: Model 1 was left unadjusted to observe the nocturia and PIR relationship visually. Model 2 included some demographic and socioeconomic factors as confounding variables. Model 3 included all possible confounding variables to resemble the patient's situation closely. Therefore, using three models allows for progressive correlation analysis between nocturia and PIR, making our experimental design more logical.

Question 5: Ensure uniformity in the fonts.

Our response: We sincerely apologize for the error. We have revised the table to ensure consistency in the text (Table 1). Additionally, we have modified the affiliations to make them more formal (line 7, line 9-12).

---

## [Decision Letter · Decision Letter 1]

3 May 2024

Association between family income to poverty ratio and nocturia in adults aged 20 years and older: a study from NHANES 2005-2010

PONE-D-24-06446R1

Dear Dr. Jia,

We’re pleased to inform you that your manuscript has been judged scientifically suitable for publication and will be formally accepted for publication once it meets all outstanding technical requirements.

Kind regards,

Sairah Hafeez Kamran, PhD

Academic Editor

PLOS ONE

Reviewers' comments:

Reviewer's Responses to Questions

**Comments to the Author**

1. If the authors have adequately addressed your comments raised in a previous round of review and you feel that this manuscript is now acceptable for publication, you may indicate that here to bypass the “Comments to the Author” section, enter your conflict of interest statement in the “Confidential to Editor” section, and submit your "Accept" recommendation.

Reviewer #3: All comments have been addressed

2. Is the manuscript technically sound, and do the data support the conclusions?

Reviewer #3: Yes

3. Has the statistical analysis been performed appropriately and rigorously? 

Reviewer #3: Yes

4. Have the authors made all data underlying the findings in their manuscript fully available?

Reviewer #3: (No Response)

5. Is the manuscript presented in an intelligible fashion and written in standard English?

Reviewer #3: Yes

6. Review Comments to the Author

Reviewer #3: ASSOCIATION BETWEEN FAMILY INCOME TO POVERTY RATIO AND NOCTURIA STATUS: A STUDY FROM NHANES 2005-2010

The manuscript titled, “Association between family income to poverty ratio and nocturia status: a study from NHANES 2005-2010”. This study examined the association between the occurrences of nocturia in adults aged 20 and above and the Poverty Income Ratio (PIR) using data from NHANES.

Further discussion on potential mechanisms underlying the observed correlation have been discussed briefly and some limitations and recommendations have been outlined. In all, comments have been addressed and hence manuscript suitable for publication.

7. PLOS authors have the option to publish the peer review history of their article (what does this mean?). If published, this will include your full peer review and any attached files.

Reviewer #3: **Yes: **PAULINE BOACHIE-ANSAH

---

## [Editor Report · Acceptance letter]

9 May 2024

PONE-D-24-06446R1 

PLOS ONE

Dear Dr. Jia, 

I'm pleased to inform you that your manuscript has been deemed suitable for publication in PLOS ONE. Congratulations! Your manuscript is now being handed over to our production team.

Kind regards, 

on behalf of

Dr. Sairah Hafeez Kamran 

Academic Editor

PLOS ONE